

# AAA+ proteases: the first line of defense against mitochondrial damage

Gautam Pareek

Cell and Molecular Biology, St. Jude Children's Research Hosp, Memphis, TN, USA

## ABSTRACT

Mitochondria play essential cellular roles in Adenosine triphosphate (ATP) synthesis, calcium homeostasis, and metabolism, but these vital processes have potentially deadly side effects. The production of the reactive oxygen species (ROS) and the aggregation of misfolded mitochondrial proteins can lead to severe mitochondrial damage and even cell death. The accumulation of mitochondrial damage is strongly implicated in aging and several incurable diseases, including neurodegenerative disorders and cancer. To oppose this, metazoans utilize a variety of quality control strategies, including the degradation of the damaged mitochondrial proteins by the mitochondrial-resident proteases of the ATPase Associated with the diverse cellular Activities (AAA+) family. This mini-review focuses on the quality control mediated by the mitochondrial-resident proteases of the AAA+ family used to combat the accumulation of damaged mitochondria and on how the failure of this mitochondrial quality control contributes to diseases.

## RATIONALE: THE CONUNDRUM OF THE DAMAGED MITOCHONDRIA AND THE ROLE OF THE SURVEILLANCE PATHWAYS

Mitochondria are indispensable for cellular life and perform a plethora of functions, including calcium homeostasis, regulation of programmed cell death, regulation of innate immunity, and stem cell fate (*Nunnari & Suomalainen, 2012*; *Bock & Tait, 2020*; *McBride, Neuspiel & Wasiak, 2006*; *Kauppila, Kauppila & Larsson, 2017*; *Chandel, 2015*; *Vasan, Werner & Chandel, 2020*). Most important of all, mitochondria produce almost all the vital cellular energy in the form of Adenosine triphosphate (ATP) by virtue of the different respiratory chain (RC) complexes embedded in the inner mitochondrial membrane (*Milenkovic et al., 2017*; *Gustafsson, Falkenberg & Larsson, 2016*; *Sousa, D'Imprima & Vonck, 2018*; *Zhao et al., 2019*). These RC complexes are involved in coupling the mitochondrial membrane potential created by the proton gradient to the synthesis of ATP. However, mitochondria face a quality control challenge during their lifetime. In addition to producing ATP, these RC complexes, mainly complexes I and III, produce a large amount of reactive oxygen species (ROS) by electron leakage, which results in the partial reduction of oxygen to form the detrimental superoxide radicals (*Quinlan et al., 2013*; *Murphy, 2009*). The excessive ROS produced by this process presents a quality control

Corresponding author
Gautam Pareek,
gautam.pareek@stjude.org

(QC) challenge for the mitochondria and the cell. The dual genetic origin of the mitochondria also poses a challenge to its homeostasis. Mitochondria harbor their own small circular DNA, which encodes for 37 genes, including 13 proteins of the RC complexes, 22 tRNAs, and two rRNAs of mitochondrial ribosomes in the human (*Hällberg & Larsson, 2014*). The vast majority of the remaining more than 1,000 mitochondrial proteins are encoded by the nuclear genome and are imported into the mitochondria by the translocase of the outer membrane 'TOM' and translocase of the inner membrane 'TIM' complex (*Pfanner, Warscheid & Wiedemann, 2019*; *Neupert & Herrmann, 2007*). Therefore, mitochondrial biogenesis, specifically the RC complexes' biogenesis, heavily relies on the coordinated expression of the nuclear and mitochondrially encoded subunits. Any perturbation in this equilibrium can lead to the accumulation of protein aggregates and eventually an adverse effect on mitochondria and the cellular health (*Youle, 2019*).

Thankfully, to combat the damage and the occurrence of ROS and misfolded protein aggregates, mitochondria have evolved a variety of surveillance pathways that are activated upon the trigger of the stress (*Rugarli & Langer, 2012*; *Pickles, Vigié & Youle, 2018*; *Roca-Portoles & Tait, 2021*; *Quiles & Gustafsson, 2020*). The most studied pathway of all is the 'mitophagy' pathway which involves the degradation of the whole mitochondria in the lysosome through a mitochondrial selective form of autophagy in extreme stress conditions (*Narendra, Walker & Youle, 2012*; *Whitworth & Pallanck, 2017*). In yeast, the mitophagy pathway is carried out by the 'Atg32' protein, which binds to an adaptor 'Atg11', and then mitochondria are recruited to and imported into the vacuole for degradation and recycling of its contents (*Kanki et al., 2009*). However, metazoans possess entirely different machinery for mitophagy, including the kinase 'Pink1' and the ubiquitin ligase 'Parkin' and several other adaptors, including P62, Optineurin, Nuclear dot protein 52 (NDP52), Human T-Cell Leukemia Virus Type I Binding Protein 1 (TAX1BP1) and Neighbor of BRCA1 gene 1 (NBR1) (*Narendra et al., 2008*; *Narendra et al., 2010*; *Jin et al., 2010*; *Lazarou et al., 2015*; *Martin, Dawson & Dawson, 2011*). Interestingly, the mutations in the genes of the mitophagy pathway, including Pink1 and Parkin, have been shown to cause Parkinson's disease in humans and genetic disruption of these genes causes locomotor and behavioral deficits and accumulation of dysfunctional mitochondria in a variety of model organisms (*Greene et al., 2003*; *Scarffe et al., 2014*; *Clark et al., 2006*; *Park et al., 2006*; *Itier et al., 2003*; *Lee et al., 2017*; *Shin et al., 2011*; *Yang et al., 2006*). The mechanistic details of this pathway have been discussed in pioneering reviews in the past and are out of scope for this review article (*Pickles, Vigié & Youle, 2018*; *Whitworth & Pallanck, 2017*; *Ge, Dawson & Dawson, 2020*).

The Mitochondrial network is dynamic; mitochondria merge by 'fusion' and separate from each other by 'fission' (*Chen & Chan, 2017*; *Friedman et al., 2011*; *Youle & van der Bliek, 2012*). The mitochondrial 'fusion' allows mitochondria to exchange their content and promotes functional complementation in the event of mitochondrial damage. In contrast, mitochondrial 'fission' facilitates the segregation of damaged mitochondria and subsequent removal by mitophagy. These processes are crucial to mitochondrial proteostasis since mutations in the components of the fission/fusion machinery have been shown to cause neurodegenerative diseases in humans, including Charcot-Marie-Tooth

Type 2A (CMT2A), peripheral neuropathy, and Dominant Optic Atrophy (DOA) (*Chan, 2020*).

The contribution of 'mitophagy' and 'mitochondrial dynamics' to mitochondrial quality control is indisputable. However, recent reports have demonstrated that these pathways cannot account for the total mitochondrial protein turnover in the cell (*Vincow et al., 2013*; *Vincow et al., 2019*). The quantitative proteomic studies performed in fruit flies and *Drosophila* S2 cells have demonstrated that the half-lives of different ribosomal proteins lie very close to each other, in agreement with the fact that the whole ribosomes are turned over as a single unit by the 'ribophagy' pathway. However, in the same study, mitochondrial proteins exhibited a wide range of half-lives that cannot be explained by a sole mitophagy-mediated mitochondrial protein degradation, suggesting more than one mechanism responsible for the turnover of mitochondria (*Vincow et al., 2013*). Subsequent studies showed that nearly 35% (one-third) of all mitochondrial protein turnover occurred through autophagy and almost 25% through the parkin-dependent mitophagy (*Vincow et al., 2019*). Additionally, utilizing mito-keima and mito-QC based reporter assays, it has been shown that the basal mitophagy flux is tissue-specific, and genetic disruption of the Pink1 or Parkin has a very modest effect on this flux (*Lee et al., 2018*; *McWilliams et al., 2016*; *Sun et al., 2017*). That being the case, the quality control mediated by the ATPase Associated with diverse cellular Activities (AAA+) serine proteases represent likely candidates to account for the majority of the mitochondrial protein turnover as they sense and degrade selective damaged mitochondrial components, henceforth, explaining the broad distribution of mitochondrial protein half-lives.

## SCOPE AND SURVEY METHODOLOGY

The mitochondrial AAA+ proteases have gained significant attention and broader interest from the scientific community from different research areas due to their role in human health and diseases. To supplement the existing information as relevant literature about AAA+ proteases has increased in the past years, this perspective review discussed the classification, structural insights, functions, and recently reported substrates of the AAA+ proteases. The relevant pieces of literature on the topic were identified using databases like Pubmed, Google Scholar, Web of Science, and ScienceDirect. Briefly, the review identified 154 relevant research articles from research labs working on AAA+ proteases across the globe. To help move the field forward, this study presents a systematic review structured to discuss (1) the need for mitochondrial QC and how AAA+ proteases supplement this need, (2) the neurodegenerative diseases associated with the AAA+ proteases, (3) what have we learned from the disease model systems, and (4) substrates of the AAA+ proteases.

## GENERAL OVERVIEW AND CLASSIFICATION OF THE AAA+ PROTEASE FAMILY

The FtsH (Filamentous temperature sensitive H)-related AAA+ proteases are present in all kingdoms of life, from bacteria to humans, and are classified into different families depending on their localization and domain organization (*Sauer & Baker, 2011*; *Quirós, Langer & López-Otín, 2015*). This family of proteases forms oligomeric complexes

that use energy from ATP hydrolysis to unfold and transport substrates to their zinc metalloprotease domain for degradation. These proteases are comprised of an axial pore formed by the oligomeric ring of the AAA+ ATPase domain. This pore size is narrow enough to block the passage of any folded protein toward an inner proteolytic chamber where proteolysis occurs (*Sauer et al., 2004*). Therefore, the energy derived from the cycles of ATP hydrolysis by the ATPase domain is utilized to induce the conformational changes in the substrate protein to unfold and transport substrates to the proteolytic cavity for degradation.

Higher eukaryotes, including humans, have five major nuclear DNA encoded $AAA^+$ proteases in the mitochondria, distinguished by their subunit composition and localization (Fig. 1A). The i-AAA (commonly known as Yme1l1) and m-AAA proteases are localized in the inner membrane, and the Lon and Clp family of proteases are matrix-localized (*Deshwal, Fiedler & Langer, 2020*). Furthermore, the catalytic domains of the m-AAA and i-AAA proteases face the matrix and the intermembrane space, respectively, to protect against the insults on both sides of the inner membrane. The m-AAA protease is classified into two forms: hetero-oligomeric complexes of the paraplegin protein and the ATPase family gene 3-like 2 (Afg3l2) protein, and the homo-oligomeric complexes of the Afg3l2 protein. For all the AAA+ proteases, the ATPase and the proteolytic subunits are contained on a single polypeptide separated by a short linker region, except the Clp family, which is composed of the proteolytic (ClpP) and ATPase (ClpX) subunits present in the two different proteins (Fig. 1A).

Recent reports based on cryo-electron microscopy have provided unique structural insights into the Yme1l1, Afg3l2, and Lon protease (*Puchades et al., 2017*; *Puchades et al., 2019*; *Shin et al., 2021*). The quaternary organization of yeast Yme1l1 protease has revealed that the six AAA+ domains form an asymmetric spiral staircase on the top of a planar symmetric hexametric protease ring (Fig. 1B) (*Puchades et al., 2017*). Notably, four of the six subunits in Yme1l1 are in the ATP-bound form; the lowest subunit has the ADP-like density in the nucleotide-binding pocket, whereas the binding pocket of one of the subunits possesses an "apo-like" nucleotide-free state. The substrate engagement in the asymmetric ring is coordinated by the two conserved tyrosine residues forming a spiral staircase around the translocating substrate by intercalating into its backbone (Fig. 1B). The sequential ATP-hydrolysis around the ring results in subsequent substrate translocation to the proteolytic domain in a stepwise fashion. The cryo-EM structure of the human truncated Afg3l2 protein comprising the ATPase and protease domain has highlighted that the basic structural features and the fundamental mechanisms of ATP hydrolysis and substrate translocation are relatively conserved between the i-AAA and m-AAA family of proteases (*Puchades et al., 2019*). However, some unique evolutionary protein sequence adaptations related to their extreme C-terminus in i-AAA and m-AAA proteases enable them to process their distinct substrates in a different microenvironment. Surprisingly, the cryo-EM structure of the human Lon protease has revealed that the substrate binding alone to the AAA+ domain is not sufficient to produce allosteric transmission to the proteolytic domain and the activation of the proteolytic chamber was only achieved in the presence of peptidomimetic inhibitor 'bortezomib' suggesting an

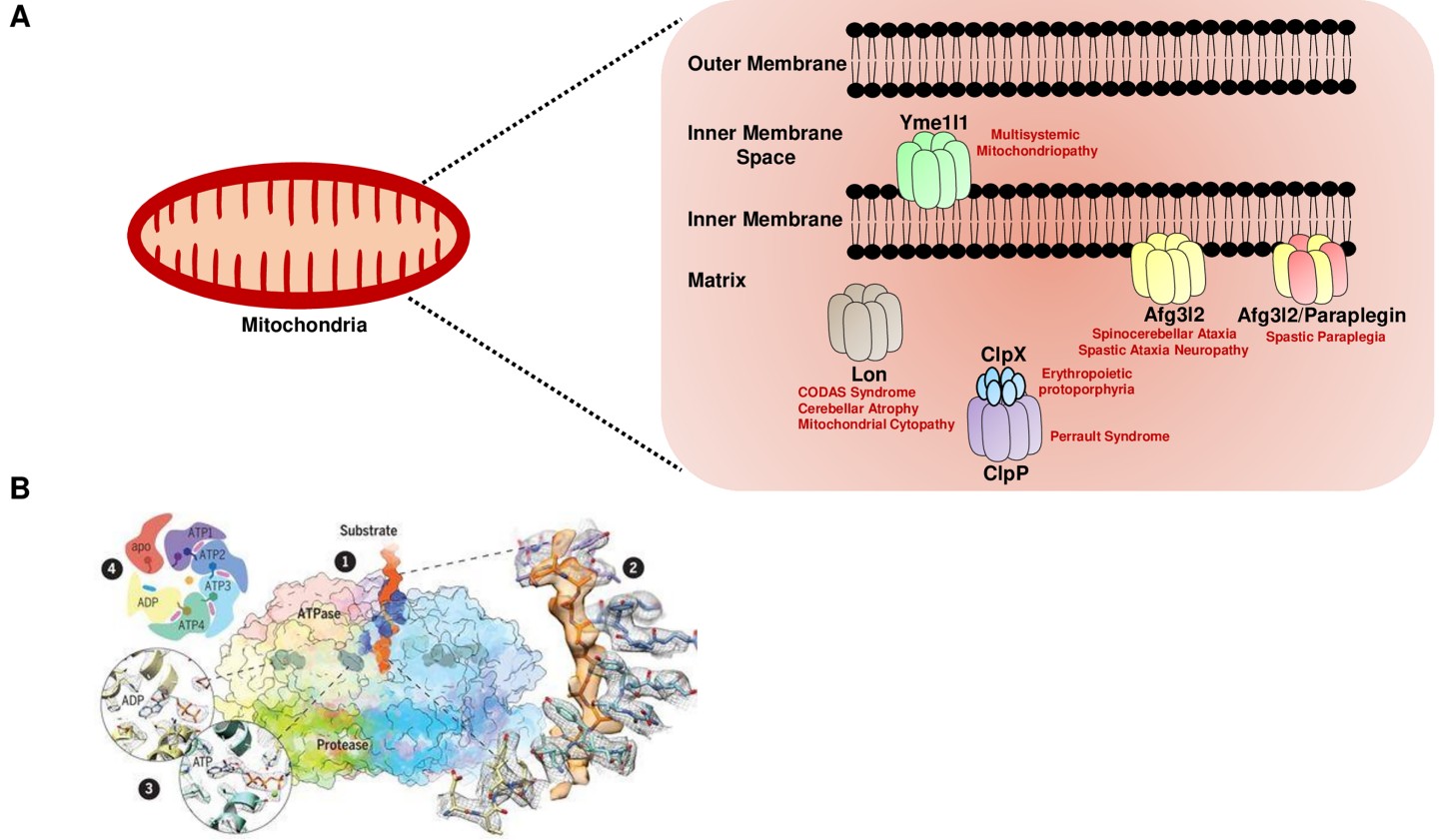

**Figure 1 The mitochondrial AAA+ protease family.** (A) The mitochondrial AAA+ proteases are classified based on their composition and sub-compartment localization. The i-AAA protease Yme1l1 is embedded in the inner membrane with active sites oriented in the intermembrane space. The m-AAA proteases are localized in the inner membrane and comprise the Afg3l2 homomers and Afg3l2-Paraplegin heteromers. The active sites of m-AAA proteases are present in the matrix. The Lon and Clp family of proteases are matrix localized. Note that for the Clp family, the protease function is carried out by the ClpP, while the ClpX protein carries out the ATP hydrolysis-mediated unfoldase function. The failure of this machinery results in devastating incurable diseases in humans. (B) The quaternary organization of the yeast Yme1 protease by cryo-EM. (1) The surface diagram depicting the ATPase domain from each subunit forming an asymmetric spiral staircase above a planar symmetric protease ring. The pore loop tyrosine residue (in blue) intercalates into the translocating substrate (in orange). The Nucleotides are in gray densities. A zoom-in view (2) highlights the cryo-EM density of a tyrosine residue from the pore loop of each subunit intercalating into the target protein, and (3) reveals the nucleotide state of the ATPase domain from subunits. (4) A cartoon showing the asymmetric organization of the Yme1 ATPase hexamer along with the nucleotide density for each subunit around the central translocating substrate. Note that the intersubunit signaling motif protrudes into the nucleotide-binding cavity of the neighboring subunit only when it is bound to the ATP. "From Structure of the mitochondrial inner membrane AAA + protease YME1 gives insight into substrate processing, Volume: 358, Issue: 6363, (DOI: 10.1126/science.aao0464). Reprinted with permission from AAAS." © 2017 American Association for the Advancement of Science.

additional level of regulation which requires the substrate-binding within the protease domain to induce the activated conformation of the domain (*Shin et al., 2021*).

The activity of all AAA+ proteases is under strict scrutiny to prevent the uncontrolled degradation of mitochondrial proteins. The precise mechanism by which the AAA+ proteases engage with their specific substrates lies in the binding to the particular sequences that serve as potential degron or degradation signals on the target protein (*Glynn, 2017*; *Steele & Glynn, 2019*; *Leonhard et al., 2000*). These degradation signals are usually highly hydrophobic and are 10–20 amino acids long (*Gur & Sauer, 2008*; *Rampello & Glynn, 2017*; *Leonhard et al., 1999*). The Yme1l1 protease recognizes its substrates

through a specific motif of amino acids, including the F-h-h-F (F = phenylalanine and h stands for any hydrophobic residue) accessible for degradation in the unfolded state but hidden in a natively folded form of the protein (*Shi, Rampello & Glynn, 2016*). The accumulation of protected intermediates from experiments involving dihydrofolate reductase and stabilizing ligand methotrexate has demonstrated that the degradation of the target protein by Yme1l1 is achieved by processive unfolding and translocation to the proteolytic chamber from the degron terminus (*Shi, Rampello & Glynn, 2016*). Furthermore, a highly similar sequence (WRFAWFP) rich in aromatic amino acids from β-galactosidase has been identified as the primary recognition site for the Lon protease (*Gur & Sauer, 2008*). The human Lon protease has been shown to preferentially recognize and degrade the unfolded proteins harboring degron signals, including the folding incompetent form of the ornithine transcarbamylase (OTC) but not the aggregated form of the malate dehydrogenase (MDH) (*Bezawork-Geleta et al., 2015*). Moreover, The peptidase specificity profile of Afg3l2 has demonstrated that this protease discriminates its potential substrates by cleaving peptide bonds adjacent to either the hydrophobic (most commonly phenylalanine) or small polar residues in the P1' position (the position adjacent to the scissile bond) (*Ding et al., 2018*). Overall, the recent structural details provide a molecular framework to understand the mechanics of the AAA+ protease machines and to appreciate their role in health and disease.

## m-AAA proteases

The m-AAA proteases are embedded in the inner membrane with their active site oriented towards the matrix. There are two different versions of m-AAA; one is composed entirely of the homoligomers of the Afg3l2 protein; the other is a hetero-oligomeric complex of the Afg3l2 and Paraplegin proteins. The heterozygous missense mutations in the *AFG3L2* gene cause spinocerebellar ataxia (SCA28) disease characterized by cerebellar dysfunction due to Purkinje cell degeneration (Fig. 1A) (*Di Bella et al., 2010*). Most of these mutations are clustered in the protease domain's highly conserved region, and one of the mutations is present at the asparagine 432 positions in the ATPase domain. All these mutations result in the loss of the Afg3l2 activity and impair the degradation/maturation of its substrates. The homozygous missense mutation in *AFG3L2* causes spastic ataxia neuropathy syndrome characterized by progressive lower limb spastic paraparesis, cerebellar atrophy, peripheral neuropathy, ptosis, dystonia, and progressive myoclonic epilepsy (Fig. 1A) (*Pierson et al., 2011*). This homozygous recessive mutation alters the conserved tyrosine at 616 positions to cysteine and impairs the homo- and hetero-oligomerization propensity of the Afg3l2 protein. The homozygous missense mutations in the *SPG7* gene (encoding Paraplegin protease) cause the hereditary spastic paraplegia (HSP) disease characterized by the progressive weakness and spasticity of the lower limbs and by the loss of upper motor neurons of the corticospinal tracts (Fig. 1A) (*Wilkinson et al., 2004*). Additionally, compound heterozygous and heterozygous mutations in the *SPG7* gene are also responsible for the progressive external ophthalmoplegia with early or progressive ataxia, dysphagia, and proximal myopathy (*Pfeffer et al., 2014*). A closer examination of these patients' skeletal muscle biopsies revealed evidence of the mosaic respiratory chain

deficiency and an increase in mitochondrial biogenesis, giving rise to 'ragged-blue' fibers. These defects were accompanied by accelerated clonal expansion of mitochondrial DNA mutations and deletions along with increased mitochondrial mass and hyper-fused mitochondria in affected individuals. The deficiency of the yeast homolog of m-AAA protease Yta10/Yta12 is viable but causes a growth defect on a nutrient source containing glycerol that requires a mitochondrial respiration (*Arlt et al., 1996*). The homozygous knockout mice of *AFG3L2* exhibited a defect in axonal development and myelination accompanied by early paraparesis and tetraparesis (*Maltecca et al., 2008*). These mice do not survive beyond day 16. The heterozygous *AFG3L2* mouse recapitulated some pathological features of SCA28, including cerebellar atrophy, dark degeneration of Purkinje cells, and mitochondrial dysfunction (*Maltecca et al., 2009*). These findings suggest that haploinsufficiency is responsible for the ataxia disease associated with mutations in the *AFG3L2* gene. The knockout of *AFG3L2* in *Drosophila* is larval lethal, and RNAi-mediated knockdown causes shortened lifespan, locomotor deficits, and severe neurological and mitochondrial morphological abnormalities (*Pareek & Pallanck, 2020*). Paraplegin deficiency in various model organisms, including *Drosophila* and mouse, is viable. Still, adults exhibited severe phenotypes, including shortened lifespan, behavioral deficits, neurodegeneration, and accumulation of dysfunctional mitochondria along with the disorganized cristae network (*Ferreirinha et al., 2004*; *Atorino et al., 2003*; *Pareek, Thomas & Pallanck, 2018*). The paraplegin deficient mice exhibited mitochondrial morphological abnormalities in synaptic terminals and distal regions of axons followed by axonal swelling, degeneration, and behavioral abnormalities. The ultrastructural morphology of the photoreceptor terminals of the *Drosophila* paraplegin knockout displayed the progressive neurodegenerative phenotype, severely swollen and dysmorphic mitochondria accompanied by altered axonal transport of mitochondria. Despite co-assembling as a heteromeric complex, the spectrum of diseases caused by mutations in the Afg3l2 and paraplegin are distinct. The interplay between these proteins and their substrates in the mitochondria is poorly understood.

The m-AAA protease Afg3l2 plays a crucial role in regulating mitochondrial translation by participating in the maturation of one of the large mitochondrial ribosomal subunits, Mrpl32 (*Nolden et al., 2005*). The Mrpl32 maturation by Afg3l2 protease is conserved in yeast, flies, mice, and human cell lines (*Pareek & Pallanck, 2020*; *Almajan et al., 2012*). The defect in the processing led to the defective ribosome assembly and the attenuation of the mitochondrial translation (Fig. 2). This subsequently caused a defect in the RC complexes comprised of the subunits encoded by the mitochondrial DNA, including complex I, III, IV, and V (*Pareek & Pallanck, 2020*). An imbalance in the stoichiometry of nuclear *vs.* mitochondrial DNA encoded RC subunits caused the accumulation of protein aggregates and activation of the mitochondrial unfolded protein response (mito-UPR) (Fig. 2). Afg3l2 deficient worms (*AFG3L2* has been confusingly referred to as *SPG7* in worms) served as one of the first models to understand the mito-UPR (*Nargund et al., 2012*). The activation of the mito-UPR restores protein homeostasis utilizing a two-pronged approach consisting of increased expression of chaperones and proteases to

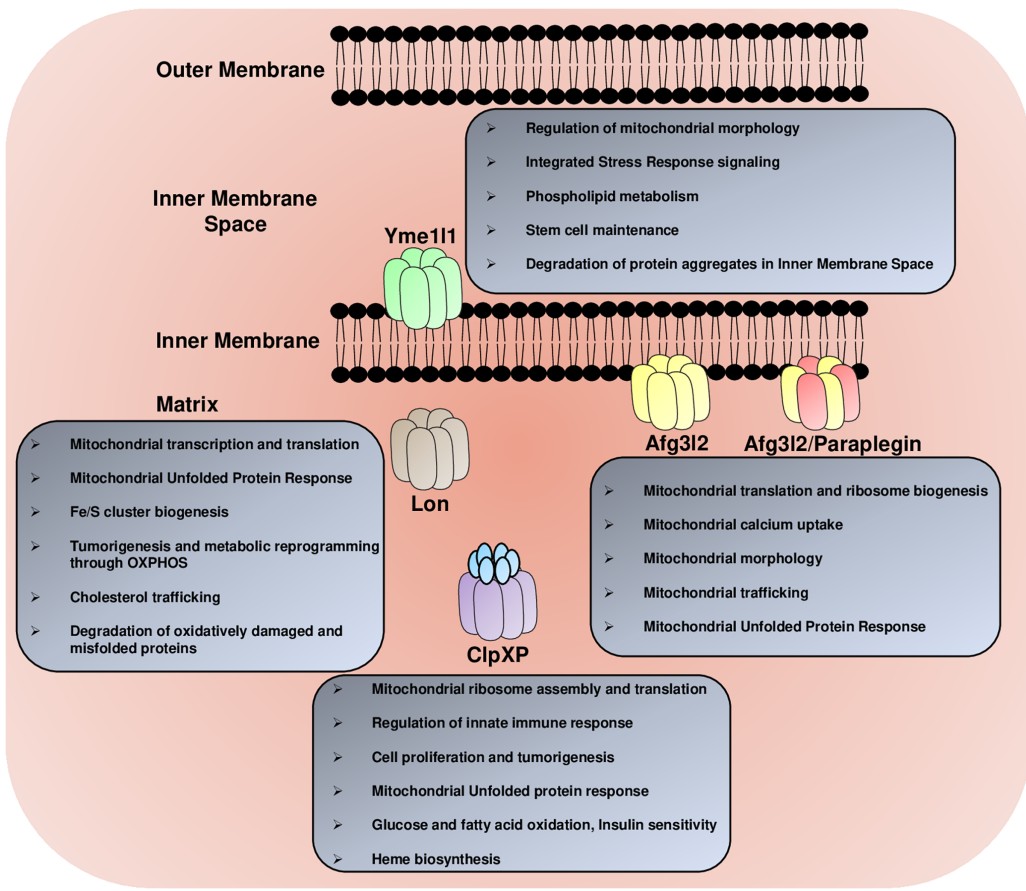

**Figure 2 The multitasking AAA+ protease family.** The mitochondrial AAA+ proteases regulate a diverse array of functions in the mitochondria, including the regulation of mitochondrial morphology, mitochondrial trafficking, glucose, fatty acids and lipid metabolism, mitochondrial calcium homeostasis, Fe/S cluster synthesis, mitochondrial transcription and translation, mitochondrial ribosome biogenesis, heme formation, innate immune response and degradation of misfolded protein aggregates.

facilitate the refolding/degradation of the misfolded proteins and phosphorylation-mediated inactivation of the cytosolic translation-initiation factor eIF2α by GCN2 kinase to attenuate cytoplasmic translation to reduce the burden on the mitochondrial protein folding machinery under stress conditions (*Baker et al., 2012*). However, some of these phenotypes, including the Mrpl32 maturation defect and activation of the mito-UPR, have not been observed in the Paraplegin deficient flies (*Pareek & Pallanck, 2020*). These recent findings suggest that the Afg3l2 homomers and the Afg3l2/Paraplegin heteromultimers have independent substrates. The result further supports this hypothesis that the *SPG7* deficient flies did not exhibit defects in the activity or abundance of several RC complexes that contain subunits encoded by the mitochondrial genome and instead showed a defect in the complex II activity, which is encoded by the nuclear genome (*Pareek, Thomas & Pallanck, 2018*).

The mitochondria also play a crucial role in calcium buffering and have a calcium uniporter complex (MCU) to uptake calcium from the cytoplasm, thereby helping regulate

the cytoplasmic calcium levels (*Baughman et al., 2011*). Recent findings have reported that the m-AAA proteases degrade a regulatory subunit of the MCU complex, known as the Essential MCU REgulator (EMRE) (*Sancak et al., 2013*; *König et al., 2016*; *Tsai et al., 2017*). The formation of the constitutively active MCU-EMRE subcomplex in the absence of another regulatory subunit, Mitochondrial Calcium Uptake Protein 1 (MICU1), caused the calcium overload in the mitochondrial matrix and triggered the opening of the mitochondrial permeability transition pore (mPTP) and necrosis mediated cell death (*Bernardi, 2013*). However, the role of the m-AAA protease in the regulation of the MCU is disputed since the depletion of the MCU failed to rescue the behavioral defects, Purkinje cell degeneration, and the neuroinflammatory response of the Afg3l2 mutants (*Patron, Sprenger & Langer, 2018*). While in contrast to this, it has also been shown using cultured Purkinje cells derived from the cerebella of newborn mice that the depolarized mitochondria in Afg3l2-deficient cells are not efficient to buffer induced $Ca^{2+}$ peaks, increasing the cytoplasmic $Ca^{2+}$ concentrations, which subsequently triggers the dark cell degeneration (*Maltecca et al., 2015*). The calcium buffering is particularly critical for the Purkinje cells as they are comprised of the highly branched dendritic arbors receiving inputs from glutamatergic stimulation of various receptors, including ionotropic α-amino-3-hydroxy-5-methyl-4-isoxazole propionic acid (AMPA) and metabotropic receptors mGluR1 and, are therefore exposed to massive and sudden $Ca^{2+}$ influx, making them susceptible to glutamate-mediated excitotoxicity. Notably, in the haploinsufficient Afg3l2 SCA28 mice, the partial genetic depletion of the metabotropic glutamate receptor mGluR1 or administration of the β-lactam antibiotic ceftriaxone, which promotes synaptic glutamate clearance, both decreased $Ca^{2+}$ influx in Purkinje cells. It also improved the ataxic phenotype (*Maltecca et al., 2015*). More future work will be required to examine this matter in greater detail to determine how sensitive the mitochondrial calcium uptake is in response to the dosages of Afg3l2 protein.

The Afg3l2 protease control several other aspects of mitochondrial biology, including mitochondrial morphology, by regulating the processing of the inner mitochondrial membrane protein Opa1 (Fig. 2) (*Ehses et al., 2009*; *Ishihara et al., 2006*). The inner mitochondrial membrane fusion success depends on the balanced stoichiometry of the large and small isoforms of the Opa1 protein (*Mishra et al., 2014*). The deficiency of the Afg3l2 m-AAA protease caused the activation of another protease Oma1 (*Ehses et al., 2009*). The enhanced Oma1 activity led to the cleavage of the Opa1 long isoform to the shorter isoform and, subsequently, mitochondrial fragmentation. The Afg3l2 protease also cleaves mitophagy protein Pink1 inside the mitochondrial matrix, and the knockdown of the protease stabilizes the smaller form of Pink1 generated after the matrix processing peptidase (MPP) cleavage (*Greene et al., 2012*; *Thomas et al., 2014*). However, the stabilization of this cleavage product did not upregulate the downstream mitophagy pathway and did not affect the parkin recruitment to the mitochondria. Depleting the Afg3l2 protease in mouse primary cortical neurons drastically impaired the anterograde transport of the mitochondria, thereby depleting the synapses out of the mitochondria (*Kondadi et al., 2014*). The underlying cause of this defect has been attributed to the tau

hyperphosphorylation by kinases, including the protein kinase A (PKA) and ERK1/2, which were activated by high ROS caused by the depletion of the Afg3l2 protease.

### i-AAA protease

The Yme1l1 is an ATP-dependent metalloprotease embedded in the inner mitochondrial membrane, with its protease domain facing the intermembrane space (referred to as i-AAA protease). The homozygous missense mutation in the *YME1L1* is associated with a multisystemic mitochondriopathy with neurological abnormalities, including intellectual disability, motor development and speech delay, and optic nerve atrophy with visual and hearing impairment (*Hartmann et al., 2016*). This mutation alters the highly conserved arginine at 149 positions within the predicted mitochondrial targeting sequence to the tryptophan. It abrogates the maturation of the Yme1l1 by MPP after the import into the mitochondria. The homozygous Yme1l1 knockout in *Drosophila* caused shortened lifespan, locomotor deficit, photoreceptor degeneration, and apoptosis-mediated cell death (*Qi et al., 2016*). These flies possessed mitochondria with disrupted cristae network, dysfunctional RC activity, and misfolded protein aggregates. The nervous system-specific knockout of *YME1L1* in mice caused microphthalmia, cataracts, progressive axonal degeneration of dorso-lateral tracts, and inflammation in the retina and spinal cords accompanied by locomotor impairment of hind limbs (*Sprenger et al., 2019*). These mice also manifested late onset RC dysfunction, loss of cristae structure, mitochondrial fragmentation, and a defect in the anterograde transport of the mitochondria.

The role of the Yme1l1 protease in regulating mitochondrial morphology is well established (Fig. 2). The Yme1l1, along with the Oma1 protease, holds a switch that dictates mitochondrial fusion/fragmentation in response to membrane depolarization and cellular bioenergetic status (*Rainbolt et al., 2016*). In conditions where membrane depolarization occurs in the presence of ATP, the Oma1 protease is degraded in a Yme1l1-dependent manner. These conditions stabilize the Opa1 short isoform generated by the Yme1l1 cleavage at the S2 site and favor mitochondrial fusion with enhanced RC activity, protection from mitophagy, and protection from apoptosis. However, when the depletion of ATP accompanies mitochondrial membrane depolarization, the Yme1l1 protease is degraded, and Oma1 is stabilized. The stabilization of Oma1 protease caused the cleavage of Opa1 at the S1 site and, subsequently, blockage in fusion, reduced RC activity, susceptibility to mitophagy, and apoptosis. Moreover, a recent study suggested that the Yme1l1 is a part of the enormous protease complex formed by the Yme1l1-PARL proteases along with the scaffold protein Stomatin-like protein 2 (SLP2). This complex SLP2–PARL–YME1L1 not only regulates the Pink1 kinase and Pgam5 phosphatase processing but also inhibits the stress-activated Oma1 protease and promotes the cell survival (*Wai et al., 2016*).

The Yme1l1 protease also participates in the integrated stress signaling (ISR) pathway during stress conditions (Fig. 2). The Yme1l1 degrades the import motor component Tim17A downstream of the stress-regulated translational attenuation induced by the eukaryotic initiation factor 2α (eIF2α) phosphorylation (*Rainbolt, Saunders & Wiseman, 2015*; *Rainbolt et al., 2013*). The decrease in Tim17A protein level attenuates the general

mitochondrial protein import and promotes the induction of the mito-UPR-associated proteostasis genes and stress-responsive genes.

Another intriguing function of Yme1l1 is related to cellular phospholipid metabolism. Yme1l1 drives the degradation of the PRELI-like protein family in the intermembrane space, including Ups1 and Ups2 in the absence of their binding partner Mdm35, a member of the twin Cx9C protein family (*Potting et al., 2010*). The Ups1 and Ups2 proteins were shown to regulate the accumulation of cardiolipin and phosphatidylethanolamine in the mitochondria (*Tamura et al., 2009*). Recent studies have also highlighted the role of Yme1l1 in metabolic reprogramming by rewiring the mitochondrial proteome under hypoxic and nutrient-deprived conditions (*MacVicar et al., 2019*). This metabolic adaptation is operated through the mTORC1-LIPIN-YME1L1 axis. The inhibition of mTORC1 activates LIPIN, a phosphatidic acid phosphatase that causes a decrease in the phosphatidylethanolamine levels in the mitochondrial membrane and subsequently activates Yme1l1. Under these conditions, the enhanced degradation of the protein translocases by activated Yme1l1 inhibits mitochondrial biogenesis. It promotes the metabolic flux of Tricarboxylic acid (TCA) intermediates towards anaplerotic biosynthetic reactions in preexisting mitochondria. These events are critical for the growth of solid tumors, such as pancreatic ductal adenocarcinomas (PDACs), for the anchorage-independent growth of the cells, and for the proliferation of cells in the presence of glutamine as a carbon source. The Yme1l1 is not only required for the development of specific cancer subtypes but also plays a role in the maintenance and self-renewal of neural stem and progenitor cells (NSPCs), and conditional deletion of the protease promotes stem cell exhaustion and pool depletion (*Wani et al., 2022*). The *YME1L1l* deletion rewired cellular metabolism in these cells, including a significant reduction in the mitochondrial fatty acid oxidation, and activated a differentiation program in NSPCs.

## Lon protease

Lon is the master AAA+ protease that protects against protein aggregation in the mitochondria by degrading the misfolded proteins. The homozygous or compound-heterozygous mutations in the Lon protease are associated with a multisystem developmental disorder known as CODAS syndrome, characterized by cerebral, ocular, dental, auricular, and skeletal anomalies (Fig. 1A) (*Strauss et al. 2015*). These mutations are clustered in the AAA+ domain, causing a defect in the protease activity. The lymphoblastoid cell lines harboring these mutant substitutions had swollen mitochondria with electron-dense inclusions, abnormal cristae structure, and a defect in the RC activity. The mutation in the proteolytic domain has also been shown to cause the atypical CODAS syndrome with cerebellar atrophy of high intensity, regression, and involuntary movement. However, the molecular basis of the defect for this mutation is not entirely understood (*Inui et al., 2017*). Additionally, mutations in the Lon protease are associated with a broad spectrum of diseases, including substitution at Pro761Leu in the proteolytic domain causes profound neurodegeneration with progressive cerebellar atrophy, hypotonia, muscle weakness, and intellectual disability (*Nimmo et al., 2019*). Interestingly, the cultured fibroblast from the affected individuals exhibited electron-dense inclusions in

the mitochondria, regular activity of the RC complexes, and glucose-repressed oxygen consumption, while the galactose and palmitic acid utilization were unaffected. Notably, these fibroblasts also had reduced pyruvate dehydrogenase (PDH) activity and elevated intracellular lactate: pyruvate ratios caused by an increase in the phosphorylated E1α subunit of PDH. A recent report has identified a recessive mutation in the N-terminal substrate recognition domain of the Lon protease at aspartate 436 to asparagine, causing a form of mitochondrial cytopathy with early onset ataxia, developmental delay, emotional outbursts, speech and swallowing difficulties, and hypotonia (*Hannah-Shmouni et al., 2019*). The muscle biopsy from patients revealed increased oxidative stress, a block in autophagy, reduced mitochondrial state 3 respiration, and intra-mitochondrial globular inclusions. The homozygous knockout of Lon is lethal in mice and *Drosophila*, while the RNAi-mediated knockdown is associated with shortened lifespan and behavioral deficits (*Pareek et al., 2018*; *Quirós et al., 2014*). Deleting the Lon protease homolog in worms (lonp-1) is viable but accompanied by disturbed mitochondrial homeostasis, ROS accumulation, impaired growth, behavior, and lifespan (*Taouktsi et al., 2022*).

In addition to the m-AAA proteases, Lon protease has also been shown to regulate the various aspects of the mitochondrial gene expression machinery (Fig. 2). The Lon protease regulates mitochondrial DNA copy number by controlling the mitochondrial transcription factor A (TFAM) levels in *Drosophila* S2 cells (*Matsushima, Goto & Kaguni, 2010*). The heterodimer of mitochondrial seryl-tRNA synthetase (SerRS2) and its paralog SLIMP protein bridges the interaction between TFAM and Lon protease, thereby promoting the TFAM degradation in S2 cells (*Picchioni et al., 2019*). While in mammalian cells, Lon protease promotes the degradation of TFAM phosphorylated by a cAMP-dependent protein kinase, which impairs its binding to DNA (*Lu et al., 2013*). Additionally, the Lon protease inhibition in fibroblast is associated with a defect in the maturation and solubility of a subset of proteins required for mitochondrial DNA maintenance and translation, including SSBP1, MTERFD3, and FASTKD2 (*Zurita Rendón & Shoubridge, 2018*). Depleting Lon in these cell lines also resulted in the loss of mitochondrial DNA, suppression of mitochondrial translation associated with impaired ribosome biogenesis, protein aggregates in the matrix, and activation of the ISR pathway. Notably, inhibition of the Lon protease in Hela cell lines caused defects in the turnover of mitochondrial pre-RNA processing nuclease MRPP3, followed by accumulation of many unprocessed mitochondrial transcripts and arrest in the mitochondrial translation (*Münch & Harper, 2016*).

The Lon protease has also been implicated in tumorigenesis, and heterozygous Lon mice are protected against colorectal cancer and skin papillomas induced by the treatment of 7,12-dimethylbenzanthracene and tetradecanoylphorbol acetate (DMBA/TPA) (*Quirós et al., 2014*). Higher expression of Lon protease is a poor prognosis marker and correlates with poor survival in human colorectal cancer and melanoma. The Lon protease-induced metabolic reprogramming, including reduction in mitochondrial oxidative phosphorylation (OXPHOS) capacity and increase in glycolysis, supports the proliferation of tumor cells and metastasis. In addition to its protease function, Lon protease also possesses chaperone properties, and this function of Lon is critical for its

anti-apoptosis activity. The increased Lon expression sequestered the P53 in the mitochondrial matrix and prevented both the transcription-dependent (by reducing the expression of P53 target genes in the nucleus) and independent (by preventing the loss of mitochondrial membrane potential and the release of apoptotic proteins cytochrome C and SMAC/Diablo in the cytosol) functions of P53 and thereby inhibited the apoptosis and promoted tumorigenesis (*Sung et al., 2018*). The high expression levels of Lon protease correlate positively in oral squamous cell carcinoma (OSCC) patients.

One of the well-established functions of the Lon protease is to degrade oxidatively damaged proteins in the mitochondria (Fig. 2). The aconitase's oxidized form was among the first reported substrates of the Lon protease (*Bota & Davies, 2002*). Subsequently, the oxidized form of several other proteins, including mitochondrial stress proteins such as Hsp78, Hsp60, Sod2, prohibitins, mitochondrial metabolic enzymes such as pyruvate dehydrogenase complex, alpha keto glutarate dehydrogenase complex, ketol acid reductoisomerase, mitochondrial ribosomal proteins Mrp20 and respiratory chain subunits such as Qcr2 (complex III), Rip1 (complex III), Cox4 (complex IV), and ATP1,2 and 7 (complex V) exhibited altered abundance in Pim1 (Lon protease homolog in yeast) deficient yeast cells (*Bayot et al., 2010*). In conjunction with the Clp protease, the Lon protease degrades subunits of the matrix exposed ROS generating arm of the complex I, including NDUFV1, NDUFV2, NDUFS1, NDUFS2, NDUFB8, and NDUFA9 in depolarized mitochondria of SH-SY5Y and HeLa cells after CCCP treatment (*Pryde, Taanman & Schapira, 2016*). The Lon protease also protects against oxidative stress in *Drosophila* (*Pomatto et al., 2017*). Notably, the sex-specific isoforms of Lon protease confer protection against hydrogen peroxide-induced stress in female flies and paraquat-induced stress in male flies. The increase in oxidative stress is also associated with protein aggregation. The Lon protease degrades several misfolded proteins and protects against protein aggregation in the human mitochondria (*Bezawork-Geleta et al., 2015*). Recently, two independent reports have suggested that the chaperone activity of Lon protease is critical to prevent aggregation of several mitochondrial proteins, including DnaJ co-chaperone (TID1/DNAJA3), succinate dehydrogenase complex flavoprotein subunit A (SDHA), ClpX and mitochondrial heat shock protein 75 kDa (TRAP1/mtHSP75) (*Matsushima et al., 2021*; *Pollecker, Sylvester & Voos, 2021*). Not only this, but Lon also forms a complex with the components of the 'TOM' and 'TIM' machinery and maintains the soluble state of newly imported proteins, and degrades the unprocessed form of mitochondrial proteins with signal sequence (*Matsushima et al., 2021*).

Almost any biological process related to mitochondria involves the regulation by the Lon protease. The Lon protease regulates the pyruvate dehydrogenase kinase 4 (PDK4) content in cardiomyocytes mitochondria and thereby controls the activity of pyruvate dehydrogenase complex and metabolic flexibility (*Crewe et al., 2017*). The ISU (Iron-sulfur cluster assembly protein) protein involved in Fe/S cluster biogenesis is also a substrate of the Lon protease when the sulfur donor Nfs1 and J-protein cochaperone Jac1 (J-type Accessory Chaperone) do not bind it (Nitrogen-Fixing Bacteria S-Like Protein) (*Ciesielski et al., 2016*; *Song, Marszalek & Craig, 2012*). The Lon protease also degrades the Steroidogenic acute regulatory protein (StAR) protein which facilitates the rate-limiting

step in steroid biosynthesis, which is the transfer of cholesterol from the outer mitochondrial membrane to the inner mitochondrial membrane (*Granot et al., 2007*). The extensive repertoire of substrates for Lon protease makes it unarguably the master of all trades in the mitochondria.

## Clp protease

The Caseinolytic mitochondrial matrix peptidase is a highly conserved member of the serine protease family residing in the mitochondrial matrix. In contrast to other proteases, which contain domains with protease and ATPase activity on the same polypeptide, the mitochondrial ClpP lacks ATP hydrolysis activity and only possesses the domain for the proteolytic activity that can digest small peptides without ATP requirement. Additionally, the ClpP, in conjunction with the chaperone and ATPase ClpX, can digest larger substrates in the matrix compartment. The ClpXP complex comprises the CLPP units arranged as a stacked heptameric ring with a proteolytic cavity flanked at both ends by ClpX (*Fei et al., 2020*). The recessive mutations in ClpP are associated with Perrault syndrome, a disease characterized by sensorineural hearing loss and ovarian dysgenesis (*Jenkinson et al., 2013*). The ClpP-deficient mouse model exhibited female and male infertility caused by disrupted spermatogenesis and ovarian follicular differentiation failures (*Gispert et al., 2013*). These mice also showed reduced lifespan, growth retardation, mild impairment in behavioral and respiratory chain activities, and activation of the Type I IFN responses through the Mitochondrial DNA–cGAS–STING signaling axis (*Torres-Odio et al., 2021*). Contrary to the mouse model, the deficiency of ClpP in fungal species Podospora anserine led to a healthy and increased lifespan, a phenotype that the expression of human ClpP can revert, demonstrating functional conservation between human and fungal ClpP (*Fischer et al., 2013*).

Over the past years, some of the substrates of ClpP protease have been reported highlighting its role in mitochondrial biology. The ClpP protease has been shown to determine the rate of mitochondrial protein synthesis and mitoribosome assembly by regulating the levels of various components of the transcription and translation machinery (Fig. 2). In the human cell culture system, the deficiency of ClpP causes the accumulation of ERAL1 (Era Like 12S Mitochondrial RNA Chaperone 1), a putative 12S rRNA chaperone, whose timely removal from the small mitochondrial ribosomal subunit is essential for the complete maturation and assembly of the mitochondrial ribosome and the normal rate of mitochondrial translation (*Szczepanowska et al., 2016*). In *Drosophila* Schneider S2 cells, the knockdown of ClpP protease caused the accumulation of leucine-rich pentatricopeptide repeat domain-containing protein 1 (LRPPRC1) followed by an increase of mitochondrial mRNAs, accumulation of some unprocessed mitochondrial transcripts, and repression of the mitochondrial translation (*Matsushima et al., 2017*).

Mitochondria are fundamental for cellular metabolism and regulate several metabolic pathways related to glucose homeostasis and fatty acid oxidation. Regarding this, the deficiency of the ClpP protease in mice is metabolically beneficial and has been associated with the lean body phenotype, increased energy expenditure, improved glucose

homeostasis, insulin resistance, protection from high fat diet-induced obesity, and protection from hepatic steatosis (*Becker et al., 2018*; *Bhaskaran et al., 2018*). However, the depletion of ClpP also caused a decline in brown adipocyte function, impaired cold-induced thermogenesis, and increased fatty acid oxidation in white adipose tissues. It remains to be seen if targeting the ClpP protease using specific inhibitors in fat tissues will help combating obesity-related disorders in humans.

In *Caenorhabditis elegans*, the ClpP homolog (CLPP-1) is required for the mito-UPR signaling (Fig. 2) (*Haynes et al., 2007*). The CLPP-1-mediated proteolysis of misfolded proteins is an early step necessary for UPR signaling in worms. However, the function of the ClpP in the UPR signaling is debated, and it has been shown in cell culture and mouse models that the ClpP is dispensable for the mito-UPR (*Seiferling et al., 2016*; *Rumyantseva, Popovic & Trifunovic, 2022*). Surprisingly, the deficiency of the ClpP protease ameliorated the symptoms of mitochondrial cardiomyopathy and mitochondrial encephalopathy caused by the tissue-specific deficiency of the mitochondrial aspartyl tRNA synthase, DARS2 (*Seiferling et al., 2016*; *Rumyantseva, Popovic & Trifunovic, 2022*; *Dogan et al., 2014*). These findings have demonstrated that the mammalian ClpP is neither required for nor regulates the UPR and introduces ClpP as a possible novel target for therapeutic intervention in mitochondrial diseases characterized by respiratory chain and mitochondrial gene expression dysfunctions.

The ClpP is overexpressed in approximately 40% of Acute Myeloid Leukemia (AML) patients (*Cole et al., 2015*). The inhibition of the ClpP protease by knockdown or by using chemical inhibitors, including A2-32-01, reduced the growth and viability of several AML cell lines with high ClpP expression. Still, it did not affect other cell lines with low levels of ClpP. The effects of the ClpP-mediated growth inhibition on AML cell lines are partly mediated by the increased amount of misfolded complex II subunits, including succinate dehydrogenase A (SDHA), impaired complex II activity, reduced oxygen consumption rates (OCR), and increased ROS (*Cole et al., 2015*). An independent report has further substantiated the role of ClpP in cancer. It has shown that ClpP is universally overexpressed in primary and metastatic human cancer, correlating with poor patient survival (*Seo et al., 2016*; *Rivadeneira et al., 2015*). Using proteomics analysis, this report showed that the ClpP forms a complex with the oncoprotein survivin and the Hsp90-like chaperone TRAP-1. Together, they regulated the solubility and function of the oxidative phosphorylation Complex II subunit succinate dehydrogenase B (SDHB). Furthermore, the inhibition of the ClpP impaired oxidative capacity by causing the accumulation of misfolded SDHB, promoted oxidative stress, and ceased critical downstream signals essential for tumor cell proliferation, invasion, and metastatic dissemination *in vivo* (*Seo et al., 2016*). Not only inhibition but the hyperactivation of ClpP by imipridones, including ONC201 and ONC212, has been shown to selectively kill the malignant cells by causing a reduction in the respiratory chain complex subunits, impairing oxidative phosphorylation and mitochondrial morphology. At the same time, these compounds do not exert any effects on non-cancer cells (*Graves et al., 2019*; *Ishizawa et al., 2019*).

The unfoldase ClpX has functions beyond its partner ClpP in eukaryotes, including its role in heme biosynthesis, which is conserved from yeast to mammalian vertebrates

(Fig. 2) (*Kardon et al., 2015*). The ClpX catalyzes the incorporation of a cofactor pyridoxal phosphate (PLP) into the ALA (5-aminolevulinic acid) synthase apoenzyme by partially unfolding it and thereby generating an active form of ALA synthase and stimulates ALA synthesis, which is the first step of heme biosynthesis (*Kardon et al., 2015*; *Kardon et al., 2020*). The morpholino-mediated knockdown of ClpX in *D. rerio*. Zebrafish caused defects in red blood cell development, indicating that the ClpX is Important for vertebrate heme biosynthesis and erythropoiesis. Additionally, the dominant mutation in ClpX has been reported that results in the pathological accumulation of the heme biosynthesis intermediate protoporphyrin IX (PPIX) in erythroid cells, causing erythropoietic protoporphyria (EPP) in the affected patients (*Yien et al., 2017*).

## FUTURE PERSPECTIVE

### What substrates and pathways are regulated by the mitochondrial AAA⁺ protease family?

The mitochondrial AAA+ family of proteases is believed to provide the first line of defense against any insults. The importance of this protease family is best exemplified by the severe neurodegenerative diseases caused by mutations in their respective genes, including Hereditary Spastic Paraplegia (HSP), SpinoCerebellar Ataxia, CODAS, and Perrault syndrome. While these proteases have been studied for decades, remarkably little is known about their precise biological roles, their substrates in the mitochondria, and the mechanisms by which their mutational inactivation causes disease in humans. It is not entirely known whether manipulating the levels of selective candidate AAA+ substrates influences the behavioral and neurodegenerative phenotypes of AAA+-deficient organisms.

### Do the mitochondrial AAA⁺ proteases degrade unfolded cytoplasmic proteins, including those involved in neurodegenerative diseases?

A common characteristic of neurodegenerative diseases is the co-occurrence of dysfunctional mitochondria and cytoplasmic protein aggregates (*Ross & Poirier, 2004*; *Ruan et al., 2020*; *Vicario et al., 2018*; *Gao et al., 2019*). Recent work in yeast and cell lines suggests that such aggregates can be imported into the mitochondria and degraded by the AAA⁺ proteases (*Ruan et al., 2017*; *Li et al., 2019*; *Hu et al., 2019*). The various protease-deficient disease models offer an excellent way to test in a system whether the aggregates associated with common neurodegenerative diseases are degraded in the mitochondria. It is tempting to speculate if the overexpression and downregulation of AAA⁺ proteases in various models of Parkinson's disease and Alzheimer's disease (strains known to develop cytoplasmic protein aggregates) will modify the behavioral phenotypes and extend the life expectancy of the organism. The recent work showing that the TDP43 levels are regulated by Lon protease in the *Drosophila* model offers a glimpse of hope that AAA+ protease might have a role in the degradation of the cytoplasmic pathological aggregates (*Wang et al., 2019*).

### Is mitochondrial translational inhibition and activation of the mito-UPR a general response to any type of mitochondrial proteotoxic stress, and how does it arise?

One of the most exciting findings from recently published work on AAA+ proteases is that the respiratory chain defects resulting from the inactivation of the proteases are caused by a global reduction of the mitochondrial translation field (*Zurita Rendón & Shoubridge, 2018*; *Münch & Harper, 2016*). While extensive work on the ER and cytoplasmic protein unfolding stress response pathways have documented how unfolded protein stress triggers cytoplasmic translational arrest, the mechanism by which the mito-UPR triggers mitochondrial translational inhibition is not entirely understood. The recently published findings lay the foundation for studies aimed to test: (1) the precise mechanism by which the mito-UPR may inhibit mitochondrial translation. In the nematode *Caenorhabditis elegans*, the mito-UPR pathway requires the transcription factor ATFS-1; the mammalian ATFS-1 equivalent has only recently been identified (*Fiorese et al., 2016*). It is unclear if the targets of the ATFS-1 homolog in the mammalian system play any role in mitochondrial translational attenuation and chaperone upregulation under proteotoxic stress; (2) whether manipulations that oppose protein misfolding can restore translation under stress conditions; and (3) whether the mitochondrial translational inhibition during proteotoxic stress is beneficial or harmful.

## ABBREVIATIONS

| | |
|---|---|
| **AAA+** | ATPase Associated with diverse cellular Activities |
| **QC** | Quality Control |
| **ATP** | Adenosine Tri Phosphate |
| **RC** | Respiratory Chain |
| **ROS** | Reactive Oxygen Species |
| **TOM** | Translocase of the Outer Membrane |
| **TIM** | Translocase of the Inner Membrane |
| **NDP52** | Nuclear dot protein 52 |
| **TAX1BP1** | Human T-Cell Leukemia Virus Type I Binding Protein 1 |
| **NBR1** | Neighbor of BRCA1 gene 1 |
| **CMT2A** | Charcot-Marie-Tooth Type 2A |
| **DOA** | Dominant Optic Atrophy |
| **FtsH** | Filamentous temperature sensitive H |
| **EM** | Electron Microscopy |
| **Afg3l2** | ATPase family gene 3-like 2 |
| **OTC** | Ornithine transarbamylase |
| **MDH** | Malate Dehydrogenase |
| **SCA** | Spinocerebellar Ataxia |
| **HSP** | Hereditary Spastic Paraplegia |
| **MCU** | Mitochondrial Calcium Uniporter |
| **EMRE** | Essential MCU Regulator |

| | |
|---|---|
| **mPTP** | mitochondrial permeability transition pore |
| **AMPA** | α-amino-3-hydroxy-5-methyl-4-isoxazolepropionic acid |
| **MPP** | Matrix Processing Peptidase |
| **ETC** | Electron Transport Chain |
| **ISR** | Integrated Stress Signaling |
| **UPR** | Unfolded Protein Response |
| **CL** | Cardiolipin |
| **PE** | Phosphatidylethanolamine |
| **PDAC** | Pancreatic Ductal Adenocarcinomas |
| **NSPC** | Neural Stem and Progenitor Cells |
| **CODAS** | Cerebral, Ocular, Dental, Auricular, and Skeletal |
| **SLP2** | Stomatin-like protein 2 |
| **ISR** | integrated stress response |
| **OSCC** | Oral Squamous Cell Carcinoma |
| **LRPPRC1** | leucine-rich pentatricopeptide repeat domain-containing protein 1 |
| **OCR** | Oxygen Consumption Rates |
| **SDH** | Succinate Dehydrogenase |
| **EPP** | Erythropoietic protoporphyria |
| **OXPHOS** | Oxidative Phosphorylation |
| **PDK4** | Pyruvate dehydrogenase kinase 4 |
| **ISU** | Iron SUlfur cluster assembly protein |
| **StAR** | Steroidogenic Acute Regulatory protein |
| **Nfs1** | Nitrogen-Fixing Bacteria S-Like Protein |
| **Jac1** | J-type Accessory Chaperone |
| **ERAL1** | Era Like 12S Mitochondrial RNA Chaperone 1 |
| **AML** | Acute Myeloid Leukemia |
| **ALA** | 5-aminolevulinic acid |
| **PPIX** | protoporphyrin IX |

### Funding

The author received no funding for this work.

### Competing Interests

The author declares that they have no competing interests.

### Author Contributions

- Gautam Pareek conceived and designed the experiments, performed the experiments, analyzed the data, prepared figures and/or tables, authored or reviewed drafts of the article, and approved the final draft.

## Data Availability

This is a literature review.

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
