# Peer review of "AAA+ proteases: the first line of defense against mitochondrial damage"

_PeerJ, doi:10.7717/peerj.14350_

## Round 0.1 · original submission · Minor Revisions

Please address minor concerns of the reviewers and amend manuscript accordingly.

Reviewer 1 ·

Basic reporting

Author needs to add the structures (be it Cryo or X-ray) for all the proteases described in the manuscript. This will hugely help the readers understand the basic mechanism of their activity. Text about the diseases and referring to cartoons, both figure 1 and 2 is redundant and oversimplified.

Experimental design

No comment

Validity of the findings

No comment

Additional comments

This review is definitely well organized and has covered the broad range of areas ranging from different AAA+proteases, their structure/activities, and the diseases that emerge from the mutations of mitochondrial AAA+ proteases. However, to be accepted in PeerJ author needs to address below:
1.) In the rationale, it is not clear why AAA+ proteases are likely candidates for mitochondrial protein turnover. Needs a better transition
2.) Line 113: define what FtsH is
3.) Two of the proteases are associated with membranes, is there any evidence that these can also be associated with other membranous organelles to control the turn over of misfolded proteins elsewhere ?
4.) Which of these proteases are encoded by mitochondrial and/or nuclear DNA? Author needs to clarify
5.) Author needs to add structures of each protein (or at least Yme1l1) because simple cartoons will not help the readers understand the quaternary organization of the proteases
6.) Lines 304-321 are completely misleading, because it reads out as if Yme1l1 is needed for the growth of solid tumors. In such cases how is the balance between turnover of misfolded proteins and cancer growth maintained
7.)Though Clp protease does not have ATP hydrolysis activity, it is interesting to see this protein falling under the category of AAA+ family. Author should clarify why this protein is an exception?
8.) Finally, the author should explain (in detail ) on how these proteases sense the presence of misfolded protein for degradation

Reviewer 2 ·

Basic reporting

No comment

Experimental design

No comment

Validity of the findings

No comment

Additional comments

In this manuscript, the author provides structural and functional insights into all the mitochondrial
AAA+ proteases, with emphasis being given to the associated human disorders. The field is of
great interest and the manuscript sheds more light on it, addressing essential questions,
concerned with the international scientific/scholarly community.

The reference list covers all the relevant literature adequately and the two figures depict all
the integral information.

The quality of written English is acceptable, but some editing regarding grammatical
inaccuracies could improve the text flow. Also, more frequent usage of punctuation could
facilitate the audience to clearly comprehend the long sentences.

Overall, I found the manuscript worthy of and appropriate for publication in PeerJ.

Minor comments

-Keep consistent, throughout the manuscript and figures, the words: AAA+ (not AAA+), m-AAA
(not m-AAA) and i-AAA (not i-AAA).
-Italicize Drosophila throughout the text and C. elegans

Annotated reviews are not available for download in order to protect the identity of reviewers who chose to remain anonymous.

·

Basic reporting

Reviewer’s report for peerj-76778

In the article entitled “AAA+ Proteases: The first line of defense against mitochondrial damage”, Dr. Gautam Pareek offers a review focusing on the quality control functions of mitochondrial AAA+ proteases and their involvement in aging and disease.

Reviewer’s comments:
This is a well-structured, interesting piece of work, in an important area of research, which, during the last decade or so, has received much attention, partly due to its relevance to metabolism efficiency and control in health and disease, which may be published in Peer Journal after a few grammatical and syntax corrections.

Generally, the English is professional, clear and unambiguous. Nevertheless, before publication, the manuscript will need some editing throughout its length, due to the presence of minor errors in the grammar and the syntax.

The publications used and proposed in the References do cover the relevant literature very adequately.

The Figures integrate an important sum of information and they should be helpful to the reader.

Experimental design

This manuscript is a review of the literature on AAA+ proteases, so, a discussion on the experimental design of the study and the validity of the findings is not relevant here. Nevertheless, the provided rationale is very clear, the material presented is carefully selected and the order of presentation seems very well thought of, while the flow of reading should facilitate comprehension of state-of-the-art information in this important area of research.

Validity of the findings

This manuscript is a review of the literature on AAA+ proteases, so, a discussion on the experimental design of the study and the validity of the findings is not relevant here. Nevertheless, the provided rationale is very clear, the material presented is carefully selected and the order of presentation seems very well thought of, while the flow of reading should facilitate comprehension of state-of-the-art information in this important area of research.

---

## Round 0.2 · accepted · Accept

All issues pointed by the reviewers were adequately addressed and the manuscript was revised accordingly. Therefore, this version is acceptable now.